# Contribution of Solar Radiation and Pollution to Parkinson’s Disease

**DOI:** 10.3390/ijerph20032254

**Published:** 2023-01-27

**Authors:** Isabella Karakis, Shaked Yarza, Yair Zlotnik, Gal Ifergane, Itai Kloog, Kineret Grant-Sasson, Lena Novack

**Affiliations:** 1Environmental Epidemiology Division, Israel Ministry of Health, Jerusalem 9446724, Israel; 2Negev Environmental Health Research Institute, Soroka University Medical Center, Beer-Sheva P.O. Box 651, Israel; 3Faculty of Health Sciences, Ben-Gurion University of the Negev, Beer-Sheva P.O. Box 653, Israel; 4Neurology Department, Soroka University Medical Center, Beer-Sheva P.O. Box 651, Israel; 5Department of Geography and Environmental Development, Faculty of Humanities and Social Sciences, Ben-Gurion University of the Negev, Beer-Sheva P.O. Box 653, Israel; 6Soroka Clinical Research Center, Soroka University Medical Center, Beer-Sheva P.O. Box 651, Israel

**Keywords:** Parkinson’s disease, solar radiation, pollution, case-control study, immigrants, adaptation

## Abstract

*Background*. Parkinson’s disease (PD) is believed to develop from epigenetic modulation of gene expression through environmental factors that accounts for up to 85% of all PD cases. The main objective of this study was to examine the association between PD onset and a cumulative exposure to potentially modifiable ambient exposures. *Methods*. The study population comprised 3343 incident PD cases and 31,324 non-PD controls in Southern Israel. The exposures were determined based on the monitoring stations and averaged per year. Their association with PD was modeled using a distributed lag non-linear model and presented as an effect of exposure to the 75th percentile as compared to the 50th percentile of each pollutant, accumulated over the span of 5 years prior to the PD. *Results*. We recorded an adverse effect of particulate matter of size ≤10 μm in diameter (PM_10_) and solar radiation (SR) with odds ratio (OR) = 1.06 (95%CI: 1.02; 1.10) and 1.23 (95%CI: 1.08; 1.39), respectively. Ozone (O_3_) was also adversely linked to PD, although with a borderline significance, OR: 1.12 (95%CI: 0.99; 1.25). Immigrants arriving in Israel after 1989 appeared to be more vulnerable to exposure to O_3_ and SR. The dose response effect of SR, non-existent for Israeli-born (OR = 0.67, 95%CI: 0.40; 1.13), moderate for immigrants before 1989 (OR = 1.17, 95%CI: 0.98; 1.40) and relatively high for new immigrants (OR = 1.25, 95%CI: 1.25; 2.38) indicates an adaptation ability to SR. *Conclusions*. Our findings supported previous reports on adverse association of PD with exposure to PM_10_ and O_3_. Additionally, we revealed a link of Parkinson’s Disease with SR that warrants an extensive analysis by research groups worldwide.

## 1. Introduction

Parkinson’s disease (PD) is the fastest growing neurological disorder in the world [1]. The disease is believed to develop from epigenetic modulation of gene expression through environmental factors that accounts for up to 85% of all PD cases [2,3,4], possibly via the mechanism of an oxidative stress [5] or neuroinflammatory pathway [6]. Due to the aging of the population and the increase in life expectancy, PD can be described as a global pandemic, and the prevalence of the disease is expected to rise within the next few years [1]. As the environmental exposures are largely modifiable, their identification, elimination, or reduction has the potential to mitigate the emerging pandemic. 

An increasing number of studies have explored the association of exposure to ambient pollution with PD development. The recent meta-analyses [7,8,9,10] indicated an adverse association between PD and nitrogen oxides (NOx), carbon monoxide (CO), nitrogen dioxide (NO_2_), ozone (O_3_) and particulate matter of sizes measuring less than 2.5 μm in diameter (PM_2.5_). Nevertheless, the accumulating findings are often contradictory. While the adverse effect of exposure to particulate matter was supported by two large studies from Canada [11,12] and Taiwan [13], the researchers from the United States [14], also in a large study population, were able to confirm these results only for a subgroup of non-smoking female participants. The adverse effect recorded in multiple studies may partially reflect a reverse causation, residual confounding, and information bias as pointed out in an umbrella review of meta-analyses by Bellou [15] and Hu et al. [10]. For instance, Palacios et al. [16,17] examined the association between exposure to PM_2.5_ and particulate matter of sizes less than 10 μm in diameter (PM_10_), with the risk of PD onset through two prospective cohort studies. Adding interactions between tobacco and caffeine consumption to the analysis revealed no impact of particles on PD development.

Similarly to particulate matter, researchers exploring exposure to NO_2_ in a large number of case–control studies conducted in the Netherlands [18], Denmark [19,20], the U.S. [14], Taiwan [13,21], Korea [22] and Switzerland [23] reached contradicting conclusions. Some of these studies did not report a positive association between exposure to NO_2_ and Parkinson’s disease [13,14,18], while others confirmed the link [19,20,21,22]. 

Among environmental factors, lifelong exposure to pesticides or farming has been suggested to be a contributing factor to PD development, as concluded by meta-analyses of dozens of reports summarized by a pooled increase of risk within the range of 11–50% in the exposed groups [15,24,25,26]. 

Multiple studies explored an association of the disease with sun-related exposures, e.g., positively related ultraviolet (UV) exposure and vitamin D. Nevertheless, the results are inconclusive. For instance, Zhou et al. reported a 25-hydroxyvitamin D insufficiency to be a risk factor for Parkinson’s disease [27], although the low vitamin D values may be a consequence of the disease itself. Findings of a nationwide study in France suggested a protective effect of UV on PD development in relatively young subjects [28]. Another study suggested a link of PD to seasonality, and specifically to excessively high temperatures [29]. 

A limited number of studies examined the relationship between ambient temperature and PD. An analysis of time-series data in Spain by Linares et al. [30] indicated a short-term increase in mortality rates and hospital admissions due to PD during heat waves (RR = 1.14, 95%CI: 1.01 1.28 and RR = 1.13, 95%CI: 1.03 1.23, respectively). Nevertheless, a group led by Rowell et al. [29] recorded a protective impact of warm temperature on PD medication purchase, showing 4.5% lower rates of PD prescriptions in July and 4.2% higher rates in January. 

The Negev region is located in the southern part of Israel and covers approximately 12,000 km^2^ of the country. In the past decade, the area has been subjected to an increasing number of heat waves and dust storms transporting dust from the Sahara Desert to the eastern Mediterranean basin, both prominent characteristics of climate change. Beer-Sheva, the main metropolis in the Negev area, accounts for over 250,000 residents and features a wide range of urban and industrial activities and transportation burden. The area is naturally abundant in solar radiation and low humidity. With the unique environment of the Negev Desert coupled with the developed medical facilities and moderate exposure to industrial pollution, the Negev location constitutes a natural lab for the investigation of climate impact on health. We hypothesize that extreme meteorological conditions and exposure to particulate matter and other pollutants in the Negev will reveal the links between PD morbidity and environment in a way that cannot be effectively captured in any other set-up. 

## 2. Materials and Methods

To estimate the contribution of environmental factors to the likelihood of developing PD, we analyzed the new PD patients residing in the Negev area who were treated at the Soroka University Medical Center (SUMC) and registered in the Clalit Health Services (CHS) HMO during 2000–2020. SUMC is a tertiary 1100-bed medical facility, making it the second largest hospital in Israel and the only one in southern Israel, serving a population of 1.25 million residents. SUMC is a part of the uniform national electronic medical records system covering all health-related data of the residents. In the analysis, we used the admission–transfer–discharge (ATD) records of the SUMC hospital system that contain clinical and demographic records on all hospitalizations in the area and all Emergency Department (ED) visits, and laboratory results. SUMC is a part of CHS HMO, which covers over 70% of the Negev population. The CHS database contains complete information on patients’ medical history and primary care records, i.e., demographic information and socio-economic status, subjects’ geocoded address, chronic and acute diagnoses, laboratory records, medications dispensed by the HMO pharmacies and visits to primary care physicians and specialists. The diagnoses in the database were recorded in the International Classification of Diseases, ninth edition (ICD-9) system. The medications were recorded by their manufacturing and generic names, along with their daily defined doses and the quantity purchased on a certain day. The Jewish or Arab-Bedouin ethnicity, gender and date of birth were defined by and reconciled between the ATD and primary physician records. 

We used the Central Bureau of Statistics (CBS) administrative data for information on socio-economic status (SES) of population by geo-statistical small census areas and accounts for factors characterizing demographical distribution, education level, quality of life, unemployment rate, poverty and retirement rate [31]. The score is stored in the range of 0–10 in the CHS HMO database.

Exposure to ambient factors was determined based on the subjects’ residence addresses. We focused on ambient air pollution (NO_2_, O_3_, PM_2.5,_ and PM_10_, SO_2_ and Solar Radiation (SR)) and meteorological conditions (temperature (°C) and relative humidity (%), recorded by the monitoring stations managed by the Ministry of Environmental Protection (MEP). The ambient environmental levels of pollutants, temperature and humidity are recorded every 5 min. We chose a year as the main temporal unit in the analysis. For each monitoring station we calculated annual averages and later, assigned those to each participant in the study. The year of PD onset was considered year “0”. The information on pollutants was assigned to the study participants based on an arithmetic average of 5 proximal monitors within a radius of 20 km from their residence. Additionally, for the purposes of sensitivity analysis, we computed the weighted averages of exposure as measured by those 5 stations using an inverse of the squared distance from the subject’s location to the monitor, i.e., Euclidian distance weighting assigning higher weight to closer monitors [32]. The weights (*w*) of the ambient values were expressed by a function decreasing with distance to the monitoring station from the subjects residence address, specifically, w=1dp, with *p* = 2, as most commonly used in this field, and *d* representing the distance. 

The quality of exposure assessment and a possible misclassification bias was assessed through comparison of cases and controls in terms of their mapping quality of the subjects’ residential addresses and a missing pattern of records for certain pollutants. The residential address accuracy was estimated by a mapping quality score ranging between 1 and 10 and categorized based on full address, neighborhood or town center. These measures were provided by the Israel CBS. Due to the possibility of discrepancy between cases and controls in terms of temporal coverage of their residential area by monitoring data for certain pollutants, we recorded the proportion of years with monitored pollutants’ levels out of the follow-up and the number of monitoring stations within a 20 km radius around the subjects’ residence. These measurements were to be accounted for in the analysis in case of discrepancies between the study groups.

### 2.1. Study Design and Definitions

The study population was selected from adults (18+) registered in CHS HMO, residing in the Southern district during 2000–2020 and having a residential address recorded in the HMO charts. The PD study group comprised all incident disease cases during 2000–2020, defined either by a diagnosis recorded in the medical charts or by a medication purchase. The former was defined by the ICD9 codes: 332 (Parkinson’s disease), 332.0 (paralysis agitans) and 331.82 (dementia with Parkinsonism). The latter condition was verified by the drug-driven algorithm proposed by Chillag-Talmor et al. using the medications listed in Table A2 (Appendix A) that classified PD patients at three certainty levels—possible, probable, and definite [33,34]. The PD status in the analysis was ascertained based on at least possible level of certainty. The onset of disease was determined by the first time the PD diagnosis was recorded or PD chronic medications were dispensed. In a sensitivity analysis, we modified the levels of certainty by adding all PD cases with recorded diagnosis to the “definite” category.

To each PD case, we individually matched up to 12 controls out of the adults in the study population (registered at the CHS HMO and residing in the Southern district). The sample size considerations were based on powering a minimal detected effect of odds ratio (OR) = 1.40. The controls were matched by ethnicity (Jew vs. Arab-Bedouin), gender and exact age in years by the time of the PD onset of the case. 

The subjects were excluded from the analysis if their HMO records indicated that a subject has moved out from the Negev area during the study period. 

### 2.2. Statistical Analysis

Continuous variables were presented by mean ± standard deviation (SD), median, minimum, maximum and interquartile range values. Categorical variables were presented as proportions out of available cases. All variables were compared between study groups using univariable conditional logistic regression, to account for the clustered observations following the matching procedure.

To adjust for an imbalance between cases in the number of controls, all analyses were weighted by the inverse probability of the controls. 

As the exposure window for PD is largely unknown, we explored various time periods for up to 10 years as being possibly relevant for the disease development. To avoid a selection bias, we preferred shorter follow-ups that were available for a larger number of subjects.

We used a distributed lag non-linear modeling approach to define the environmental factors over the span of the follow-up period. The analysis was performed using a *dlnm* package, introduced by Gasparrini [35,36]. Specifically, the model framework consists of a cross-basis, a bi-dimensional space of functions specifying the relationship between the predictor and the lags. In this study, we developed cross-basis metrics using natural cubic splines for environmental predictors and a polynomial function defining the lags space with three degrees of freedom (df) for modeling exposure of five lags (and df = 5 for models with 10 lags). The parameters were chosen based on Akaike information criteria (AIC).

Transformed pollution and meteorological variables were further added as independent factors in a conditional logistic regression with PD as a dependent variable; clusters defined by the matched groups of cases and controls and inverse probability weights were assigned to all observations. The point estimates of association of PD with environmental factors were expressed as OR along with the 95% confidence intervals (CI) around ORs. The main effect of exposure was measured as exposure to pollutants at the 75th percentile over the span of exposure window, as compared to the median levels of pollutants. This difference in exposure (75th vs. 50th percentile) was chosen to reflect actual differences among the study participants. In addition to the three parameters used for matching, the ORs were adjusted to socioeconomic and immigration status, as well as an indicator of missing environmental levels, i.e., proportion of years with monitored pollutants’ levels out of the follow-up and the number of monitoring stations within the 20 km radius around the subject’s residence. In the main analysis we used a single-pollutant approach. Assuming interactions between environmental factors, we repeated the main analysis with a consecutive addition of temperature, relative humidity and pollutants that could potentially confound the association at study.

To reveal possible vulnerable sub-populations, we stratified the analysis by sex, age groups, Bedouin-Arab vs. Jewish ethnicity, socio-economic status and immigration status. Sensitivity analyses were performed for three levels of certainty. 

Additionally, we repeated the analysis for the conditions that would be the closest to the main analysis. Specifically, we explored whether PD cases were at higher odds to be exposed to pollutants in their 4th quartile (above 75th percentile), as compared to controls. We used conditional logistic regression and included the same covariates as in the main analysis.

All analyses were performed using SAS 9.4 (SAS Institute, Cary, NC, USA) and R 4.2, R studio R Core Team (2022). Vienna, Austria. 

## 3. Results

The study population comprised 34,667 subjects. It included 3343 incident PD cases identified during the study period and meeting the study requirements to whom we could match at least one non-PD control. The number of controls was 31,324 subjects, with half of the cases matched to 10 or more controls and only 5% with 1–6 controls.

The majority of the population were males (56.0%); 4.3% were of Arab-Bedouin origin and others were Jewish. Approximately a quarter of the study participants immigrated to Israel with the latest wave in the 1990s (Table 1). The subjects were on average 73.7 years old at the time of their index PD case onset. The distribution of chronic morbidities among the study participants was typical to their age. 

Environmental factors are described in Table 2. Pollution levels averaged over the study period of 2000–2020 are not excessive, except for PM_10,_ with half of the days measured beyond 39.8 µg/m^3^ featuring a semi-arid location of the study. Temperatures reached an annual average of 24.9 °C. 

As per the study protocol, all subjects had a residential address recorded in the database (Table A2, Appendix A). For approximately 80% of them, the address was recorded in full or up to the street level. In other cases, neighborhood coordinates or geographic centers of the towns were used instead. Cases were characterized by lower percentage of time in follow-up covered by the monitoring of PM_2.5_, temperature and SR, as compared to controls (49.5% vs. 56.6%, 84.5% vs. 86.6% and 79.5% vs. 81.4%, respectively).

An exposure window of 5 years was chosen for the main analysis in the study, owing to a substantially larger proportion of the study participants with this window available for the analysis—53.6%, as opposed to the 10 years, fully monitored only for 22.9% of the study population. The effect of environmental factors in a form of ORs is demonstrated in Figure 1. Specifically, ORs express an incremental impact of a hypothetical 5-year long cumulative exposure to the 75th percentile of each pollutant as opposed to their median levels. The values for the 75th and 50th percentiles used in these models correspond to the ones presented in Table 2. All estimates were adjusted to demographic factors and indicators of missing patterns of environmental measurements. Most pollutants were likely to be adversely associated with PD. Specifically, the PM_10_ and solar radiation were independently associated with Parkinson’s disease with ORs 1.06 (95%CI: 1.02; 1.10) and 1.23 (95%CI: 1.08; 1.39), respectively. Ozone was also adversely linked to the PD, although with a borderline significance, OR: 1.12 (95%CI: 0.99; 1.25). Conversely, NO_2_ presented with a protective effect, suggesting lower exposure levels to this pollutant among PD cases (OR: 0.94 (95%CI: 0.90; 0.98)). Furthermore, PD cases were likely to be exposed to lower ambient temperatures 5 years prior the disease onset (OR: 0.90 (95%: 0.81; 1.00)). These cumulative effects and their incremental estimates over the span of 5 years prior to the PD onset are presented graphically in Figure 2. 

### 3.1. Subgroup Analyses

We further examined a possible impact of pollutants in five demographic strata (Figure 3). In addition to the demographic factors and missingness pattern of pollutants’ monitored values, all the models accounted for temperature that varied between cases and controls (Figure 1). Here, we present the OR estimates only for the outstanding groups, while the estimates of their comparators are captured in Figure 3. The stratified findings indicated a higher impact of PM_10_ and SR on females, featured by OR: 2.44 (95%CI: 1.42; 4.21) and OR: 1.30 (95%CI: 1.05; 1.60), respectively. Likewise, age was likely to modify the effect of PM_10_ and SR, featuring younger patients (<70 years old) with higher OR estimates, i.e., OR: 1.95 (95%CI: 0.99; 3.85) and OR: 1.28 (95%CI: 0.99; 1.65), respectively. Additionally, older patients (85 years+) presented with estimates of high magnitude for O_3_ (OR: 2.63 (95%CI: 1.41; 4.88)) and SR (OR: 1.92 (95%CI: 0.75; 4.93), although not significant)). Immigrants from the latest wave of 1989 and later were likely to be more impacted by exposure to O_3_ and SR (OR: 1.43 (95%CI: 1.05; 1.95) and OR: 1.73 (95%CI: 1.25; 2.38), respectively). Non-immigrants presented an increased impact of SO_2_ (OR: 1.54 (95%CI: 1.07; 2.20)). Stratification by groups of SES, demonstrated effects of higher magnitude for PM_10_, PM_2.5_ and SO_2_ for a subgroup of subjects at the lowest SES level (with OR higher than 4.4 for exposure to particulate matter and OR: 1.46 for SO_2_). The second level of SES was more impacted by O_3_ (OR: 1.52 (95%CI: 1.25; 1.86) as compared to other groups.

NO_2_ demonstrating an overall protective impact on PD (Figure 1) onset changed its direction to a borderline significant effect among patients at second SES level (OR: 1.07 (95%CI: 0.98; 1.17)).

### 3.2. Sensitivity Analyses

The main analysis was based on all PD cases ascertained through diagnosis and medication, which included approximately 20% of cases at the lowest level of certainty (“possible”) (Table 1). Figure A1 (Appendix A) presents the main analysis stratified by cases’ definition at possible, probable and definite levels of certainty. Strata comprising only definite cases of PD tended towards higher point estimates of association for most of the environmental factors. These include O_3_, PM_2.5_ and SR. The OR estimate for PM_10_ was not affected by PD certainty and remained constant within the range of 1.04–1.07. NO_2_ levels were no longer protective in their association with definite PD, as compared to a seemingly protective effect obtained by the main model for all cases.

There were multiple associations recorded within the environmental factors associated with PD, e.g., NO_2_ with O_3_, with SO_2_, humidity, temperature and solar radiation (data not shown). As a part of a sensitivity analysis, the primary models were also adjusted to these factors, in order to account for their possible mutual impact. As displayed in Figure A2 (Appendix A), the main associations of PD with environment remained largely unchanged except for the widened confidence intervals around the point estimates. Controlling for temperature and humidity seemed to enhance the magnitude of the adverse association with O_3_, SO_2_ and SR.

The sensitivity analysis exploring an association between exposure to the highest quartile and PD (Table A3, Appendix A) supported the findings for the solar radiation. The magnitude of association with O_3_ and PM_10_ decreased and became nonsignificant. 

## 4. Discussion

The main objective of this study was to examine the association between PD onset and a cumulative exposure to potentially modifiable ambient exposures. The main analysis demonstrated an adverse effect of PM_10_, SR and O_3_ on PD, independent of basic demographic factors such as ethnicity, gender, age, SES, immigration status and quality of ambient exposure monitoring (O_3_, at a borderline significance). These associations were robust also when controlled for other environmental exposures and became stronger for PD cases ascertained with a “definite” level of certainty.

The current investigation benefited from taking place in a unique setup of one hospital providing tertiary care to the residents of a large geographical area. This enabled a close follow-up of study participants and granted the study a population-based quality characterized by minimal referral bias and virtually no patients lost to follow-up. 

The association of *PD with PM_10_* was recorded for cumulative exposure to the 75th percentile of this pollutant as compared to median, resulting in OR = 1.06 (95%CI: 1.02; 1.10). Findings numerically similar to these have been demonstrated in the Nurses’ Health Study (NHS), which included 115,767 female nurses with prospectively identified 508 incidents of PD cases during a period of 2 years [17]. However, a difference observed within this cohort for women exposed to the upper quartile of PM_10_ versus its lower quartile was not statistically significant (RR =1.03, 95%CI: 0.78, 1.37). The same group of researchers investigated another large cohort of men comprising the Health Professionals Follow-up Study (HPFS) [16], and concluded there was no association of PD with PM_10_ following a 16-year follow-up (RR = 0.85, 95%CI: 0.63, 1.15). In fact, studies that demonstrated an adverse association of a higher magnitude either originated from more polluted regions, such as a district in China (HR = 1.38, 95%CI: 1.12, 1.85 for an increase of an interquartile range (IQR)) [37], or resulted from a case–control approach, or both, as in a study from Taiwan (OR = 1.35, 95%CI: 1.12, 1.62 for an increase of 20 μg/m) [13]. Remarkably, in the aforementioned studies, the inspected difference in PM_10_ was at least twice higher (IQR) than the one inspected in our study (75th versus 50th percentile), which we believed was closer to the actual difference expected in real life. The area covered by our project, on the other hand, has been progressively more exposed to dust storms, the main composite of which is PM_10_ particles. Thus, our findings are in line with the results of case–control studies in areas heavily exposed to PM_10_.

With respect to *ozone*, to the best of our knowledge, the only study reporting significant results was a study by Zhao et al. [38]. It used data from the 2001 Canadian Census Health and Environment Cohort and included 3.5 million adults with 8500 deaths due to Parkinson’s. The authors concluded that an increase of one IQR in ozone averaged over 10 years of exposure was associated with an increased mortality due to the disease (Hazard Ratio = 1.09, 95%CI: 1.04, 1.14) [38]. With that being said, PD-related mortality investigated by Zhou et al. might be considered a clinically different outcome from the incident PD that is at the focus of our study. The other published reports were from North America [12,39] and Taiwan [13] with moderate adverse effects but underpowered, and a study from Korea [40] indicating a protective but far from significant impact. As ozone is known to increase with temperature, the inconclusive direction of its association with PD onset may stem from the different geographical locations of study populations and insufficient adjustment for temperature fluctuations.

The *solar radiation* component has been largely under-investigated; therefore, no reference can be set to align our findings with reports by others. The adverse association of SR with PD onset recorded in the current analysis is noteworthy but warrants more research worldwide. Likewise, speculating on possible pathophysiological mechanisms is possibly too early prior to any consensus on its statistical association being reached. 

Contrary to other reports, PD cases in our study were likely to be less exposed to *NO_2_* and *heat*. Although the former association was not confirmed within a stratum of “definite” PD cases, our findings are not in line with other studies reporting NO_2_ as a risk factor. This could be attributed to the rigorous adjustment approach used in the current analysis or an interaction with heat particularly high in the study area. 

We observed a negative association of PD with ambient temperature (OR = 0.90, 95%CI: 0.81; 1.00). There are no reports supporting or refuting these findings in relation to PD development, warranting more investigation of the possibility and direction of this link. 

Our *stratified analysis* resulted in several outstanding findings.
Based on the study findings, patients developing PD at relatively young age (<70 years) are likely to be more exposed to PM_10_, as compared to their controls. Patients diagnosed substantially later (85+) tend to be more exposed to O_3_. This observation might be related to different etiology of the disease and/or behavioral patterns featuring subjects younger than 70 and older than 85. For instance, Parkinson’s Disease may be more impacted by a genetic composite in patients diagnosed at a younger age, which might be accompanied by triggers different from older patients. Inspection of group younger than 60 at PD onset revealed an even stronger link with PM_10_, characterized by OR = 3.89 (95%CI: 1.18; 12.76), as compared to OR point estimates of 1.96, 1.62 and 1.35 for groups younger than 70, 70–85 and 85+, respectively. The findings suggest PM_10_ being a possible risk factor for an early PD onset.Another observation concerns the new immigrants (arriving in Israel in 1989 and later) who appear to be more vulnerable to exposure to O_3_ and solar radiation. It is important to mention that the vast majority of the new immigrants in the study population are from the former USSR [41]. This population is used to a significantly colder climate characterized by lower ozone and solar radiation levels; therefore, it can be considered naïve to the semi-arid conditions of the Negev desert region. On the other hand, the dose response effect of SR, i.e., non-existent for Israeli-born participants (OR = 0.67, 95%CI: 0.40; 1.13), moderate for immigrants before 1989 (OR = 1.17, 95%CI: 0.98; 1.40) and relatively high for new immigrants (OR = 1.25, 95%CI: 1.25; 2.38) indicates the adaptation abilities to solar radiation of the population in general.Subjects at the lowest SES were more likely to be impacted by SO_2_, PM_10_ and PM_2.5_. Low socioeconomic status has been established as one of the most important risk factors for many health outcomes [42,43]. It should be noted that 12.0% of the lowest SES strata are subjects of Bedouin-Arab origin, a group that comprises just 4.3% of the study group. Approximately a quarter of the Bedouin-Arabs subjects reside in temporary tents and shacks [44] that cannot be hermetically sealed from dust storms (PM_10_) or anthropogenic pollution (PM_2.5_ and SO_2_) when a subject’s home is located in close proximity of a road or industrial zone. Unprotected households combined with hazardous work exposures, poor diet, and other factors characterizing a low SES stratum [43] can potentially increase the susceptibility of the population.

Results of a sensitivity analysis testing an association between exposure in the highest quartile of a pollutant and PD supported the findings for SR but not for other pollutants. The inconsistency possibly resulted from the basic difference between the two models, whereas the dlnm function is exploring a non-linear effect of continuous values of pollutants and model in the sensitivity analysis and of a binary exposure. 

### Limitations

The current investigation presents several limitations. 

For instance, we had to limit the exposure window by 5 years despite the underlying assumption that PD is a result of long-term exposures. Nonetheless, exposures within 5 years did not vary from exposures in the 6–10 years prior to disease onset, among subjects for whom this window period was available. Therefore, we considered the 5-year window to be a valid proxy for a longer period. As a result, we were able to perform the analysis on a larger population with a minimal possibility of a selection bias. 

In attempt to be inclusive and maximally represent the wide spectrum of PD patients, the study population included all patients for whom the disease was ascertained, even at the lowest (possible) level of certainty. The population of PD cases may, therefore, include non-PD patients (e.g., Vascular parkinsonism), as indicated by a certain morbidity profile of the study subjects. With that being said, the possible misclassification of disease is expected to attenuate the point estimates of the association and therefore represents a conservative approach in the analysis. 

The study results are prone to misclassification bias due to inaccuracy in exposure assessment by monitoring stations. However, the misclassification was nondifferential in relation to the study groups for most of the pollutants; therefore, it could potentially only underestimate the magnitude of an effect. To mitigate the impact of a possible differential misclassification, we controlled for the discrepancies throughout the analyses. 

It is necessary to acknowledge that the research was carried out in a single geographical area and did not incorporate a diverse range of regions, which may be considered a limitation and potentially hinder the generalizability of the study.

## 5. Conclusions

This population-based investigation largely supported the findings of other studies with comparable methodology. Specifically, we reaffirmed an adverse impact of cumulative exposure to PM_10_ and O_3_ on Parkinson’s Disease onset, the later at borderline significance. Additionally, we revealed a link of Parkinson’s disease with solar radiation that has not been reported before, and, therefore, warrants a more extensive analysis by research groups in various geographical locations.

## Figures and Tables

**Figure 1 ijerph-20-02254-f001:**
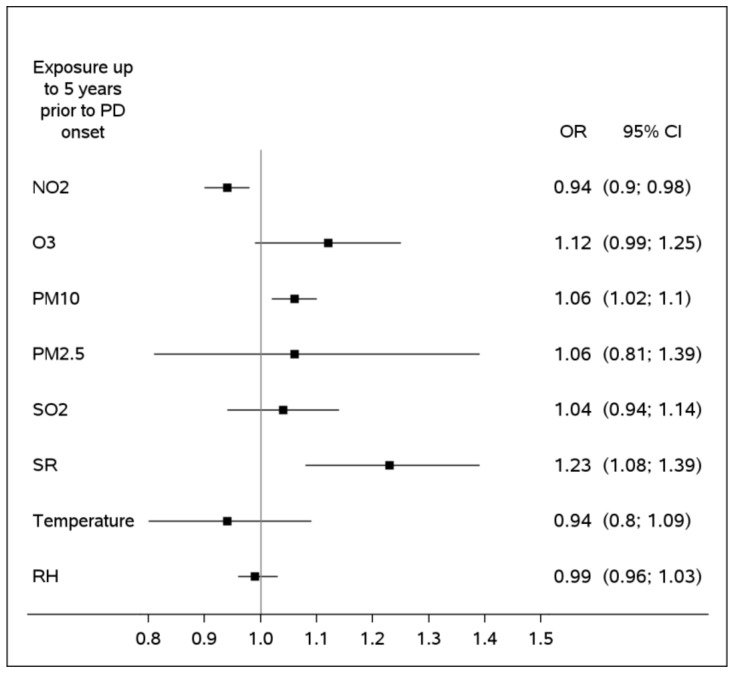
Cumulative association of Parkinson’s disease with exposure to the 75th percentile of exposure as compared to median pollutants’ levels, over 5 years. All estimates are adjusted to ethnicity, gender and age (by design) and socioeconomic status, immigration status, proportion of years in follow-up with monitored pollutants’ levels out of total years in follow-up, mapping quality score in 3 categories (full address up to street level/neighborhood/town center) and number of monitoring stations within a 20 km radius around the subject’s residence (by modeling).

**Figure 2 ijerph-20-02254-f002:**
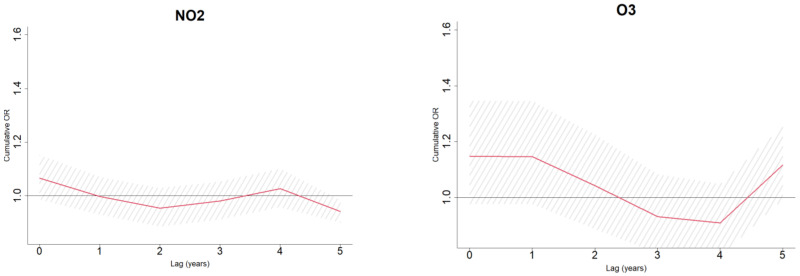
Graphical presentation of the association of Parkinson’s disease with cumulative exposure to the 75th percentile of exposure as compared to median pollutants’ levels. The red lines show the cumulative effect of a pollutant and the shaded area around the red line, represents a 95% confidence of the effect.

**Figure 3 ijerph-20-02254-f003:**
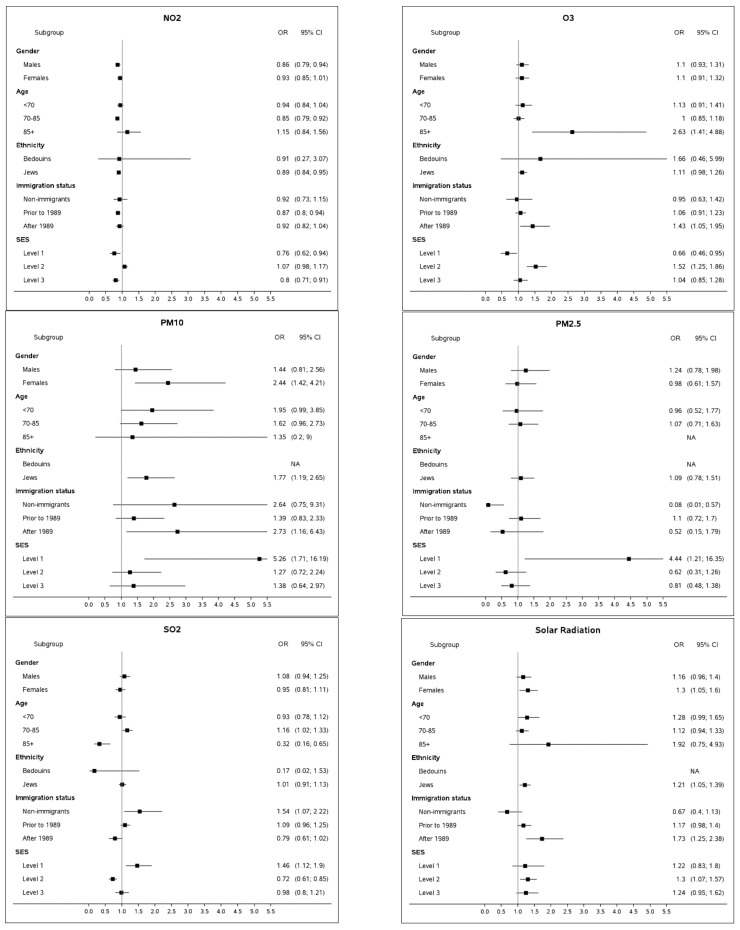
Stratified analysis of cumulative association of Parkinson’s disease with exposure to the 75th percentile of exposure as compared to median pollutants’ levels, over 5 years. All estimates are adjusted to ethnicity, gender and age (by design) and socioeconomic status, immigration status, proportion of years in follow-up with monitored pollutants’ levels out of total years in follow-up, mapping quality score in three categories (full address up to street level/neighborhood/town center) and number of monitoring stations within a 20 km radius around the subject’s residence and temperature during the same exposure window (by modeling).

**Table 1 ijerph-20-02254-t001:** Demographical and clinical characteristics of the study population.

Subjects’ Characteristics	Parkinson’s Disease Cases (N = 3343)	Controls ^2^ (N = 31,324)	*p*-Value
Case definition, % (n/N) Diagnosis alone Medication alone Diagnosis and medication Level of certainty in Parkinson’s disease ^1^, % (n/N) Possible Probable Definite	24.7 (826/3343) 41.6 (1390/3343) 33.7 (1127/3343) 20.1 (673/3343) 4.9 (164/3343) 75.0 (2506/3343)	---- ----	
Years of follow-up prior to PD onset year Mean ± SD (n) Median Min; Max	8.7 ± 5.7 (3343) 8.0 1.0; 21.0	8.9 ± 5.7 (31,324) 8.0 1.0; 21.0	0.326
*Demographical characteristics*			
Male, % (n/N)	55.8 (1865/3343)	55.6 (19,442/31,324)	----
Age at diagnosis of the case, years Mean ± SD (n) Median Min; Max	73.7 ± 10.2 (3343) 75.4 21.3; 108.7	73.7 ± 10.0 (31,324) 75.3 20.9; 108.9	----
Bedouin, % (n/N)	4.7 (157/3343)	4.2 (1318/31,324)	----
Immigration status, % (n/N) Immigrated before 1989, % (n/N) Immigrated in 1989 or later, % (n/N)	64.1 (2143/3343) 26.0 (870/3343)	61.9 (19,376/31,324) 24.6 (7703/31,324)	<0.001
Socioeconomic Status (0–10) Mean ± SD (n) Median Min; Max Grouped, % (n/N) 0–3 4–5 6+	4.9 ± 1.7 (3343) 5.0 0; 10.0 16.4 (548/3343) 53.0 (1772/3343) 30.6 (1023/3343)	4.7 ± 1.7 (31,324) 4.0 0; 9.0 28.3 (8873/31,324) 40.5 (12,680/31,324) 31.2 (9771/31,324)	<0.001 <0.001
*Medical history prior to PD diagnosis of the case*, % (n/N)			
Myocardial Infarction Congestive Heart Failure Peripheral Vascular Disease Cerebrovascular Disease, % (n/N) Dementia Chronic Pulmonary Disease Rheumatologic Disease Peptic Ulcer Disease Liver Disease Diabetes Mellitus Hemiplegia or Paraplegia Renal Disease Cancer	13.2 (440/3343) 7.1 (238/3343) 7.8 (261/3343) 22.5 (753/3343) 3.7 (124/3343) 11.0 (368/3343) 1.9 (62/3343) 5.5 (185/3343) 5.5 (185/3343) 24.5 (819/3343) 5.4 (181/3343) 13.9 (465/3343) 9.7 (324/3343)	12.8 (3997/31,324) 8.4 (2635/31,324) 7.0 (2201/31,324) 13.6 (4254/31,324) 1.8 (568/31,324) 12.2 (3816/31,324) 1.2 (381/31,324) 5.0 (1566/31,324) 5.4 (1690/31,324) 18.6 (5828/31,324) 3.9 (1208/31,324) 11.6 (3626/31,324) 13.2 (4130/31,324)	0.335 0.008 0.025 <0.001 <0.001 0.011 <0.001 0.044 0.119 <0.001 <0.001 <0.001 <0.001

^1^ Level of certainty in definition of PD was ascertained using the algorithm established by Chillag-Talmor et al. as described in the methodology section, in combination with ICD-9 diagnoses of PD. Specifically, PD cases verified by a diagnosis were assigned the “definite” level of certainty. ^2^ There were on average 9.4 ± 1.7 controls for each case, with a median of 10, a minimal count of 2 and a maximal of.

**Table 2 ijerph-20-02254-t002:** Ambient annual exposure factors.

Ambient Annual Levels During the Year of PD Onset	Parkinson’s Disease Cases (N = 3343)	Controls (N = 31,324)	Q1 ^1^	Q2 ^1^	Q3 ^1^	IQR ^1^
NO_2_, ppb Mean ± SD (n) Median Min; Max	9.0 ± 1.8 (3308) 9.5 3.8; 16.0	8.9 ± 1.9 (31,059) 9.2 1.1; 17.0	7.6	9.3	10.1	2.5
O_3_, ppm Mean ± SD (n) Median Min; Max	33.1 ± 4.7 (3233) 32.7 26.3; 82.5	32.6 ± 4.5 (30,221) 31.9 26.3; 82.5	29.5	32.1	35.5	6
PM_10_, µg/m^3^ Mean ± SD (n) Median Min; Max	39.7 ± 2.2 (3036) 39.8 33.4; 46.3	39.8 ± 2.4 (28,296) 39.8 33.4; 46.3	38.2	39.8	40.5	2.3
PM_2.5_ levels, µg/m^3^ Mean ± SD (n) Median Min; Max	19.6 ± 3.3 (1879) 19.9 12.1; 26.8	19.3 ± 3.4 (19,748) 19.8 12.1; 26.8	18.2	19.8	21.6	3.4
SO_2_, ppm Mean ± SD (n) Median Min; Max	2.4 ± 1.2 (3279) 2.1 0.3; 11.5	2.4 ± 1.3 (30,777) 2.1 0.2; 11.5	1.7	2.1	2.8	1.1
RH, % Mean ± SD (n) Median Min; Max	68.1 ± 5.7 (2833) 71.1 22.0; 78.8	68.1 ± 5.7 (26,432) 69.9 22.0; 78.8	64.0	69.9	71.8	7.8
Temperature, °C Mean ± SD (n) MedianMin; Max	19.6 ± 1.4 (3050) 19.5 14.3; 24.9	19.5 ± 1.4 (28,636) 19.5 14.3; 24.8	19.3	19.5	20.0	0.7
SR, W/m^3^ Mean ± SD (n) Median Min; Max	32.9 ± 4.6 (3025) 32.9 25.1; 41.6	32.2 ± 4.7 (28,308) 31.1 23.6; 44.8	27.6	31.1	36.7	9.1

^1^ Q1 is assigned to the 25th percentile, Q2—to median ad Q3 to the 75th percentile. IQR—interquartile range—is defined as Q3 minus Q2.

## Data Availability

The data sharing is not allowed by the current IRB approval. It can be available for other researchers upon request and only if approved by the local IRB committee.

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
