# Peer review of "Contribution of Solar Radiation and Pollution to Parkinson’s Disease"

_ijerph, 2023, doi:10.3390/ijerph20032254_

Round 1
Reviewer 1 Report
This is well written study. The novel aspect of this study is that it provides an interesting role of exposure to solar radiation, temperature and relative humidity with onset of PD. Following are my observations about this manuscript:
1. The title of the manuscript is very non-specific. Please make it specific.
2. I am not sure how useful it is to take a control population with significant medical history in this study. Can't healthy population be taken as control?
3. These types of studies can easily become victim to complicated statistical analysis. The statistical analysis done in this study is complicated and needs a great deal of simplification so that a clinician can understand the methodology and exposure-effect associations.
Author Response
January 23, 2023
Dear reviewer,
We are grateful for your time spent reading the manuscript and for your suggestions. We addressed each of the comments in a point-by-point manner, as outlined below.
Sincerely,
Lena Novack, on behalf of the research team
Reviewer 1:
This is well written study. The novel aspect of this study is that it provides an interesting role of exposure to solar radiation, temperature and relative humidity with onset of PD. Following are my observations about this manuscript:
- The title of the manuscript is very non-specific. Please make it specific.
-Thank you for pointing this out. Following the reviewer’s comment, we revised the title to include “solar radiation and pollution” in contributors to Parkinson’s Disease.
- I am not sure how useful it is to take a control population with significant medical history in this study. Can't healthy population be taken as control?
- The population of controls is expected to be maximally similar to the population of cases in terms of their socioeconomic status, genetic phenotype, medical services and have identical time to develop the disease as their controls. The later condition warranted matching by age, which came along with similar comorbidities. We believe that our matching was successful, judging by the nominal comparison of proportions (although frequently statistically significant owing to the large sample size).
- These types of studies can easily become victim to complicated statistical analysis. The statistical analysis done in this study is complicated and needs a great deal of simplification so that a clinician can understand the methodology and exposure-effect associations.
- We could not agree more with the reviewer, the message should be clear to clinicians who are not supposed to understand the specific statistical methodology. On the other hand, we had to address the audience of environmental epidemiologists who would be require those details to provide their own judgment of the research validity. We therefore, chose to compromise by leaving the details, but simplifying the results interpretation. Specifically, we revised the text on page 10 (lines 230-232) describing the main results.
Reviewer 2 Report
The impact of the environment in the context of Parkinson's Disease (PD) is an issue which is not sufficiently described in the literature. The work by Karakis et al has however certain issue which could be developed:
1. In the introduction authors should present more widely the various pathways leading to PD among which should be mentioned the neuroinflammatory one - Ref.
Platelet-to-lymphocyte ratio and neutrophil-tolymphocyte ratio may reflect differences in PD and MSA-P neuroinflammation patterns. Neurol Neurochir Pol. 2022;56(2):148-155. doi: 10.5603/PJNNS.a2022.0014. Epub 2022 Feb 4. PMID: 35118638.
2. Authors should also refer to pharmacotherapy of the patients possibly resulting in parkinsonism.
3. Due to the limitations of the study I believe it would be more suitable to present the issue as "Contribution of environment to parkinsonism". Elaborating on this issue in the context of PD due to the limitations of the study is jeopardized. In this context authors could also refer to previous works regarding coexistant environmental factors in the pathogenesis of other parkinsonisms as atypical parkinsonisms.
4. The fact that patients were evaluated in one region should also be considered as a limitation.
5. The study lacks a graphical summarizing overview.
Author Response
Dear reviewer,
We are grateful for your time spent reading the manuscript and for your suggestions. We addressed each of the comments in a point-by-point manner, as outlined below.
Sincerely,
Lena Novack, on behalf of the research team
Reviewer 2:
The impact of the environment in the context of Parkinson's Disease (PD) is an issue which is not sufficiently described in the literature. The work by Karakis et al has however certain issue which could be developed:
- In the introduction authors should present more widely the various pathways leading to PD among which should be mentioned the neuroinflammatory one - Ref.
Platelet-to-lymphocyte ratio and neutrophil-tolymphocyte ratio may reflect differences in PD and MSA-P neuroinflammation patterns. Neurol Neurochir Pol. 2022;56(2):148-155. doi: 10.5603/PJNNS.a2022.0014. Epub 2022 Feb 4. PMID: 35118638.
- Thank you for this addition. We revised the introduction to mention a neuroinflammation as a possible pathway.
- Authors should also refer to pharmacotherapy of the patients possibly resulting in parkinsonism.
- Due to the limitations of the study I believe it would be more suitable to present the issue as "Contribution of environment to parkinsonism". Elaborating on this issue in the context of PD due to the limitations of the study is jeopardized. In this context authors could also refer to previous works regarding coexistant environmental factors in the pathogenesis of other parkinsonisms as atypical parkinsonisms.
- Thank you for pointing this out. Here we will respond to the 2 items above. The case definition is of Parkinson's Disease (either physician assigned diagnosis or use of specific anti-parkinson therapy). This protocol for identifying PD patients was used in the past by our group and others (Chillag-Talmor, 2011 (https://www.ncbi.nlm.nih.gov/pubmed/22772304) and 2013 ( https://www.ncbi.nlm.nih.gov/pubmed/23939255)). The algorithm we used was previously established as a protocol identifying PD patients in a clinical database.
- We acknowledge that the clinical diagnosis of PD is sometimes not accurate, yet it is part of every large scale epidemiological study. This fact will be acknowledged in the discussion section as a limitation.
- The fact that patients were evaluated in one region should also be considered as a limitation.
- we agree with the reviewer. Performing the study in one geographical region may potentially limit its generalizability. We introduced this point in the limitations section (lines 390-392).
- The study lacks a graphical summarizing overview.
The graphical summary has been included in the submission. Below please see the graph attached in the word version of our response. We ensured that the figure appears also as a part of the revised version.

Round 2
Reviewer 1 Report
None
Reviewer 2 Report
I do not have further comments.